# Systematic Review of Diagnostic Tools and Interventions for Sarcopenia

**DOI:** 10.3390/healthcare10020199

**Published:** 2022-01-20

**Authors:** Moon Joo Cheong, Yeonseok Kang, Sungchul Kim, Hyung Won Kang

**Affiliations:** 1Rare Diseases Integrative Treatment Research Institute, Wonkwang University Jangheung Integrative Medical Hospital, Iksan 59338, Korea; sasayayoou@naver.com; 2Department of Medical History, College of Korean Medicine, Wonkwang University, Iksan 54538, Korea; yeonkang@wku.ac.kr; 3Department of Neuropsychiatry of Korean Medicine & Inam Neuroscience Research Center, Wonkwang University Sanbon Hospital, Gunpo City 15865, Korea; kscndl@hanmail.net

**Keywords:** complex exercise, aging, aerobic exercise, diagnostic criteria, sarcopenia

## Abstract

Diagnosis of rare incurable diseases is important. Specific evaluation methods and standards for sarcopenia differ according to each sarcopenia-related medical association. This study aimed to identify the tools that are currently used to diagnose sarcopenia and to systematically review various interventions for sarcopenia. We intended to provide basic information to help establish standard diagnostic and therapeutic methods for sarcopenia. We collected and analyzed published journal articles, including gray literature and dissertations, from 11 domestic and international databases. The search terms were “sarcopenia/sarcopenic”, “combined (complex/circuit) exercise”, “resistance (muscle) exercise”, and “aerobic exercise”. The tools used for sarcopenia diagnosis were inconsistent across the studies. Circuit exercise combined with aerobic exercise and strength training was the most common intervention method, followed by strength training and aerobic exercise. We identified several diagnostic and evaluation criteria across the articles. Essentially, this systematic review confirms the importance of diagnostic criteria for sarcopenia and compares interventions. Hopefully, the criteria for the diagnosis and evaluation of sarcopenia will become clear in the future. In addition, the results of this study may provide basic information for rehabilitation treatment for rare and incurable diseases.

## 1. Introduction

Sarcopenia is a newly introduced geriatric disease that was assigned a new disease classification code by the World Health Organization in October 2016 [1]. Sarcopenia is diagnosed when the skeletal muscle index, which indicates muscle mass, is below the standard value and when there are difficulties in daily living activities due to factors such as decreased grip strength or walking speed [2]. The factors that cause sarcopenia are classified as primary (aging) and secondary (drugs or diseases) [3]. In general, when diagnosing sarcopenia, three factors—muscle mass, muscle strength, and muscle function or performance—are evaluated; the standards are different for each classification [4]. Therefore, the result of the diagnosis varies depending on the criteria used for diagnosis. Early diagnosis of sarcopenia is important because the condition is rare and incurable; moreover, to the best of our knowledge, no treatment has been developed for sarcopenia [5,6], and an appropriate intervention can be used only when the diagnosis is clear. Further, relief and improvement in treatment compliance can be achieved in patients only when an accurate diagnosis is made. Therefore, it is necessary to explore a standard diagnostic method regarding sarcopenia for older adults and to investigate the literature for treatment options as well [7].

The purpose of this study was to systematically and comprehensively review existing research in order to confirm the types and standards of diagnostic tools concerning sarcopenia. In addition, for older adults, we reviewed various types of intervention methods that are used currently in clinical practice for sarcopenia.

## 2. Materials and Methods

### 2.1. Protocol and Registration

This systematic review (SR) was registered at the Research Registry (number reviewregistry1199) on 2 August 2021, (https://www.researchregistry.com/browse-the-registry#registryofsystematicreviewsmeta-analyses/registryofsystematicreviewsmeta-analysesdetails/6107e715382eb3001e4ace21/ (accessed on 2 August 2021)). 

### 2.2. Information Sources

The following 11 domestic and international databases were used by two researchers to conduct this study: five Korean language databases—Oriental Medicine Advanced Searching Integrated System, Korean Studies Information Service System, Research Information Service System, Korean Medical Database, and Korea Citation Index—and six English language databases—MEDLINE via PubMed, EMBASE via Elsevier, the Allied and Complementary Medicine Database via Elton B. Stephens Company (EBSCO), the Cochrane Central Register of Controlled Trials, the Cumulative Index to Nursing and Allied Health Literature via EBSCO, and PsycARTICLES via ProQuest. There were no language restrictions. This study was conducted from 1 September 2020 to 1 October 2020. The publication period was set to 10 years from January 2010 to August 2020.

### 2.3. Search Strategy

Two researchers independently searched the studies. We used Medical Subject Headings (MeSH) terms in multiple combinations including “sarcopenia/sarcopenic”, “combined (complex/circuit) exercise”, “resistance (muscle) exercise”, “aerobic exercise”, and “randomized controlled trial”.

### 2.4. Data Synthesis Strategy

Two researchers independently performed the data synthesis. Researchers screened titles and abstracts and obtained full reports for all titles that appeared to meet the eligibility criteria. In case of any uncertainty, the researchers discussed whether the studies met the inclusion criteria. All excluded studies were documented with reasons for exclusion. Any disagreement was resolved through consensus building.

### 2.5. Eligibility Criteria

Studies involving non-human participants were excluded from this review; that was the sole exclusion criterion. The inclusion criteria were based on the Patient/Participants/Population/Problem, Intervention, Comparison, Outcome, and Study design approach [8] and are presented below.

#### 2.5.1. Types of Participants

We included studies of individuals with sarcopenia based on diagnostic criteria that included the minimum diagnosis of muscle quantity or quality. However, because the cutoff values were based on different ethnic groups [9,10], no exact values were required, as long as the studies followed a diagnostic criterion that was supported by authoritative evidence. There were no restrictions in terms of sex, race, or age of the participants.

#### 2.5.2. Types of Interventions and Comparators

Intervention methods involved complex (combined or circuit) exercises, including aerobic exercise training, resistance exercise training, and nutrition education and dietary intake but excluding psychological interventions. The placebo was used as comparators.

#### 2.5.3. Types of Outcome Measures

Primary outcomes included measurements of body compositions, such as appendicular skeletal muscle mass (ASM), skeletal muscle mass index (SMI), percentage of body fat, fat-free mass, total fat-free mass, appendicular fat-free mass (AFFM), muscle mass index, leg lean mass, and total fat mass.

#### 2.5.4. Types of Study Design

We selected only randomized controlled trials (RCTs) for network meta-analysis. Studies using inappropriate random sequence generation methods, such as alternate allocation, were excluded.

### 2.6. Study Selection

Two researchers independently performed the selection process and compared their results to ensure that no studies were omitted. Any disagreement was resolved by discussion or by seeking advice from an external advisor. The searched studies were organized using Endnote X 9.3 reference management software (Clarivate Korea, Seoul, Korea), and duplicate results were removed. The literature selection process was presented as a Preferred Reporting Items for Systematic reviews and Meta-Analysis (PRISMA) flow diagram (Figure 1) and was reported according to the PRISMA_2020_checklist.

### 2.7. Data Extraction

Two independent researchers used a standardized data collection form to perform and cross-check the data extraction. Discrepancies were resolved through discussion with other researchers. The coding [author, year, article type, screening tools (diagnosis tools), intervention types, number of participants] was based on a previous meta-analysis of the effects of non-pharmacological interventions on sarcopenia [11,12].

### 2.8. Risk of Bias Assessment in the Included Studies

Two independent researchers assessed the studies in accordance with the Cochrane risk-of-bias (RoB) tool [13], and disagreements were resolved by seeking advice from an external advisor. The outcomes from the quality assessment of the studies were entered into the Review Manager software (version 5.3; RevMan 5.3; Cochrane, London, UK) according to the quality assessment judgment criteria.

### 2.9. Ethics and Dissemination

Ethical approval was not required because the data used in this SR did not involve individual personal patient data and there were no concerns regarding privacy. The results will be disseminated by publication of the manuscript in a peer-reviewed journal or by presentation at a relevant conference.

## 3. Results

### 3.1. Study Selection

From January 2010 to August 2020 (a period of 10 years), there were 8916 articles in domestic and foreign online databases identified using the following keywords: “sarcopenia”, “complex (combined) exercise”, and “circuit exercise”. A total of 10,166 manuscripts were searched offline, including 1250 gray papers. The studies that were suitable for the SR and meta-analysis were selected as follows: First, 6509 duplicate manuscripts, 540 pamphlets, and 2918 studies on non-human subjects were excluded, and 199 manuscripts were selected. Second, 91 articles that included participants who were not diagnosed with sarcopenia were excluded. Third, 59 papers that were not RCTs or that did not clearly indicate participation, intervention, comparator, and outcome frameworks were excluded. Finally, among the remaining 49 studies, 37 articles were included in our SR analysis, after excluding 12 non-full text articles. The literature selection process was reported in accordance with the PRISMA guidelines (Figure 1). The authors (year), sarcopenia diagnostic tool, study design, intervention, and primary outcomes of the studies are described in Table 1.

### 3.2. Sarcopenia Screening Measure(s)

Various diagnostic tools, including the European Working Group on Sarcopenia in Older People (EWGSOP) and Asian Working Group for Sarcopenia (AWGS) diagnostic criteria, have been used to diagnose sarcopenia. The results of our SR revealed that nine diagnostic tools for sarcopenia—the AWGS and EWGSOP diagnostic criteria; body mass index (BMI); body fat; ASM; muscle index; SMI; hand grip; trunk skeletal muscle mass; and AFFM—were used in the 37 studies. Moreover, there were cases where grip strength and gait speed were classified according to the AWGS and EWGSOP diagnostic criteria and sub-criteria. However, there was no unified diagnostic tool used, and the diagnostic criteria varied for each tool. For BMI as the diagnostic criterion, the fat index was 35% in four studies or 32% or 30% in other studies. Concerning the ASM tool, some studies used a cutoff of <5.4 kg, whereas others used a cutoff of 5.14 kg. Some studies had also used a skeletal mass of <32.5% as the cutoff value. Regarding SMI, some studies diagnosed sarcopenia based on a diagnostic criterion of <27.6% SMI, and some studies separately analyzed men and women who were diagnosed using <37% and 28% SMI, respectively. The above results can be confirmed in Table 1. 

### 3.3. Characteristics of Participants

Among the 37 studies, 16 were based on obesity and sarcopenia; 21 studies included both men and women, whereas 16 of the selected studies focused on women only. 

### 3.4. Outcomes

Among the selected studies, in terms of primary outcomes, 25 and 10 studies measured body composition and fitness function, respectively. Other studies measured the physiological index; degree of inflammation; blood component biomarkers; interleukin 6; secreted protein, acidic and rich in cysteine; macrophage migration inhibitory factor; and insulin-like growth factor-1. Furthermore, the fitness function and body compositions included an evaluation of grip strength and gait speed (for fitness function) as well as muscle strength, fat, body fat, and lean musculoskeletal index (for body composition).

### 3.5. Interventions

Among the 37 selected studies, 16 studies used complex exercise, 11 used resistance training, and 1 used aerobic exercise as interventions for sarcopenia. Two studies used nutritional intake as an intervention method, whereas four studies used multiple interventions involving exercise, nutritional intake, and education simultaneously. Other studies reported the use of acupuncture and self-regulated training. Regarding the total number of sessions and the length of intervention, most sessions were conducted three times per week for 12 weeks. Other studies conducted sessions three times per week for 16 weeks, three times per week for 15 weeks, or five times per week for 12 weeks. Finally, we included study results that were both positively effective and statistically significant (Table 1).

### 3.6. Risk-of-Bias (RoB)

RoB assessment was performed for the 37 included studies using Review Manager 5.4.1 (Figure 2 and Figure 3).

#### 3.6.1. Selection Bias

##### Random Sequence Generation

Five studies were rated to have a “high” RoB for using arbitrary methods, whereas eight studies were rated to have an “unclear” RoB because randomization methods were not reported. Twenty studies mentioned the use of a random number, and four studies mentioned the use of computer-generated random numbers; accordingly, they were rated to have a “low” RoB.

##### Allocation Concealment

Five studies were rated to have a “high” RoB because the participants were arbitrarily assigned, whereas 19 studies were rated to have an “unclear” RoB because there was no mention of allocation concealment. The remaining 13 studies were rated to have a “low” RoB because they used a third-party organization for randomization control.

#### 3.6.2. Performance Bias

One study was rated to have a “high” RoB because its participants were aware of the training they were undergoing. Twenty-one studies were rated to have an “unclear” RoB because they did not mention blinding of participants or personnel. The remaining 15 studies were rated to have a “low” RoB because they used control-group training.

#### 3.6.3. Detection Bias

One study was rated to have a “high” RoB because the assessor of the outcome was not blinded, whereas 31 studies were rated to have an “unclear” RoB because no blinding of the outcome assessor was mentioned. The remaining five studies were rated to have a “low” RoB because they involved blinding of the outcome assessment.

#### 3.6.4. Attrition Bias

Regarding incomplete data reporting, 12 studies reported that there were no missing data, whereas six studies not only reported missing data but also described the reasons for the missing data. Therefore, these studies were rated to have a “low” RoB. Nineteen studies had missing data but did not present clear reasons for the missing data; thus, these studies were rated to have an “unclear” RoB.

#### 3.6.5. Reporting Bias

Thirty-six studies were rated as having an “unclear” RoB because they did not state whether the study was conducted in accordance with the corresponding protocol. One study was rated to have a “low” RoB because it reported outcomes in accordance with the published protocol.

## 4. Discussion

### 4.1. Summary of Evidence

Diagnosis is the most important step when treating a disease. In particular, in the case of rare and incurable diseases, a clear diagnosis helps in determining appropriate treatment methods. In addition, it enables medical staff and caregivers to attend to problems that may arise in the future and provides a sense of security by enhancing the understanding of the disease. However, the results of our SR revealed that the diagnostic tools for sarcopenia were inconsistent across studies. Moreover, even if the same diagnostic tool was used in different studies, the diagnostic criteria were unclear. Despite the fact that the AWGS has published diagnostic criteria for sarcopenia in Asians [51], only few studies have used them. The non-systematic and inconsistent application of these criteria highlights the need for a consensus regarding sarcopenia diagnosis [52]. Furthermore, some studies used dual-energy X-ray absorptiometry scans, which allow for an objective evaluation of muscle mass [53]. However, this tool was not approved by the New Health Technology Assessment Committee before 2017; after being included in the category of restrictive health technology in 2017, it is currently under review [54]. At present, the diagnosis and treatment of sarcopenia have advanced, despite difficulties in accurately measuring muscle mass and quality in clinical practice. Notably, most studies have been conducted in women [55], suggesting that sarcopenia has sex specificity. This finding is in agreement with the finding of So in 2016 [14], who reported a higher risk of sarcopenia in women than in men. The clinical intervention was circuit exercise combining alternating resistance training and aerobic exercise; resistance training was used to increase muscle strength and aerobic exercise to decrease fat. This is consistent with the recent findings reported by Yin et al. in 2020 [12], who stated that the use of circuit exercise was increasing worldwide as an intervention for sarcopenia and other diseases in older adults. In addition, Rodrigues-Krause et al. in 2021 [56] stated that physical interventions for older adults should be structured and include aerobic and strength training, while ensuring that these adults can enjoy performing them by themselves [37]. This may be the basis for the fact that the interventions used in the selected studies mainly aimed to decrease fat using aerobic exercise such as dancing. However, it is not possible to confirm the optimal intervention for comparative advantages. To this end, we propose a study be conducted to confirm priorities of interventions through comparative analyses of future interventions. In addition, strength, muscle mass, fat, body fat, and gait performance were mainly used as tools to measure the effectiveness of the intervention, but there was no consistency in the use of diagnostic tools. There was a commonality between measuring body composition and measuring physical strength, but the reference points were not consistent. This result supports the necessity to discuss not only the consistency of diagnostic tools but also the uniformity of outcome measurements because the evaluation criteria for sarcopenia are diverse.

Finally, in South Korea, 560 adults over the age of 65 were followed for 6 years, and the risk of death was 4.1 times higher in male sarcopenia patients with insufficient muscle mass and strength. Slow walking increased the risk of death five times. There are many reports that cancer patients with sarcopenia have a shorter survival time and a worse prognosis, including recurrence. However, in the case of Korea, sarcopenia is not classified as a disease yet, and prevention or diagnosis is not clear. Therefore, if a clinician can suspect sarcopenia when discussing the ambiguity of the sarcopenia diagnostic tool and treating the elderly in clinical practice, it will be possible to provide various options to the patient. Therefore, the results of this study are expected to provide various clinical advantages for future sarcopenia research.

### 4.2. Limitations and Future Lines

The limitations of this study are as follows.

First, as this study is a systematic review study on sarcopenia diagnostic tools and interventions, quantitative numerical comparison of intervention methods could not be performed. Therefore, in future research, in order to analyze the effectiveness of interventions applied to sarcopenia, it is suggested to conduct a meta-analysis and a network meta-analysis in which comparative priority of intervention effects is possible.

Second, this study collected and analyzed only non-pharmacological and non-invasive interventional method studies. Therefore, pharmacological and invasive intervention methods could not be confirmed. For future research, we propose a study that can explore all interventions applied to sarcopenia.

## 5. Conclusions

We found that first, in general, sarcopenia was diagnosed by evaluating three factors: muscle mass, muscle strength, and muscle function or performance. However, different diagnostic and evaluation criteria were used across studies. Second, among the intervention methods for sarcopenia, the most used method was circuit exercise that combined aerobic and strength exercise. Third, the majority of the study participants were middle-aged women with obesity. According to our findings, no clear diagnostic tool was identified for the diagnosis of sarcopenia as various criteria were applied. In addition, the main intervention involved the application of complex exercise combining aerobic and strength exercise and there were many studies wherein participants with sarcopenia were obese.

## Figures and Tables

**Figure 1 healthcare-10-00199-f001:**
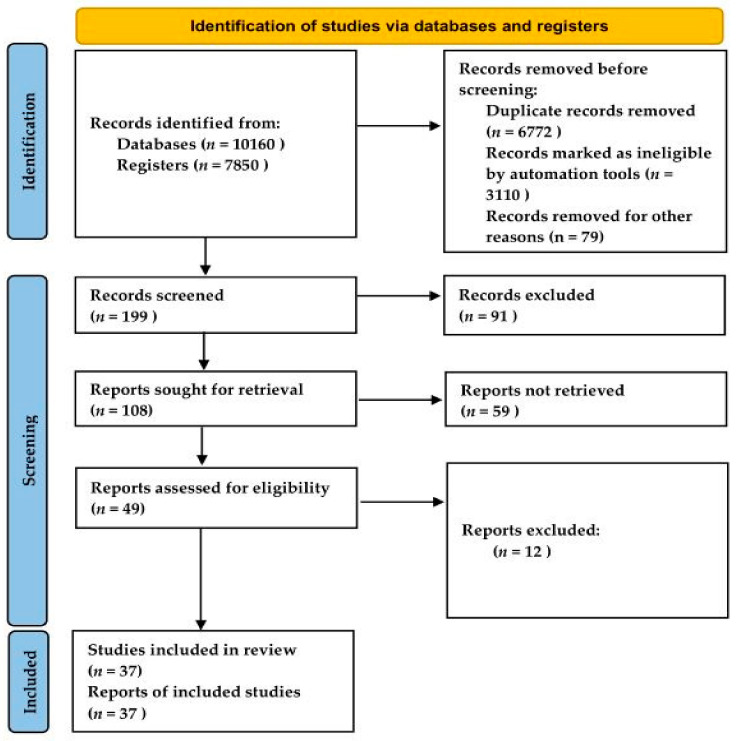
PRISMA flow chart of Paper search for systematic review on diagnostic tools and intervention methods for Sacopenia.

**Figure 2 healthcare-10-00199-f002:**
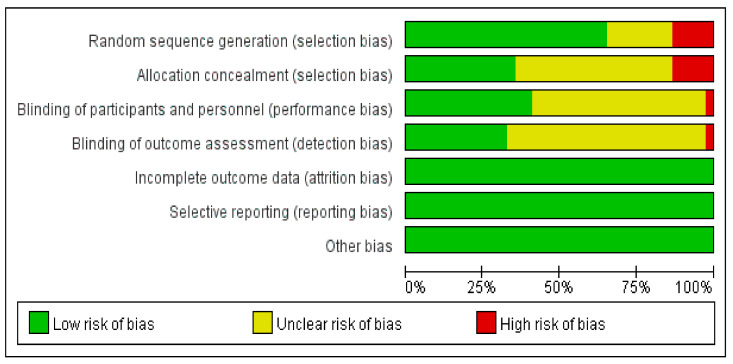
Risk-of-bias graph of systematic review on diagnostic tools and intervention methods for Sacopenia.

**Figure 3 healthcare-10-00199-f003:**
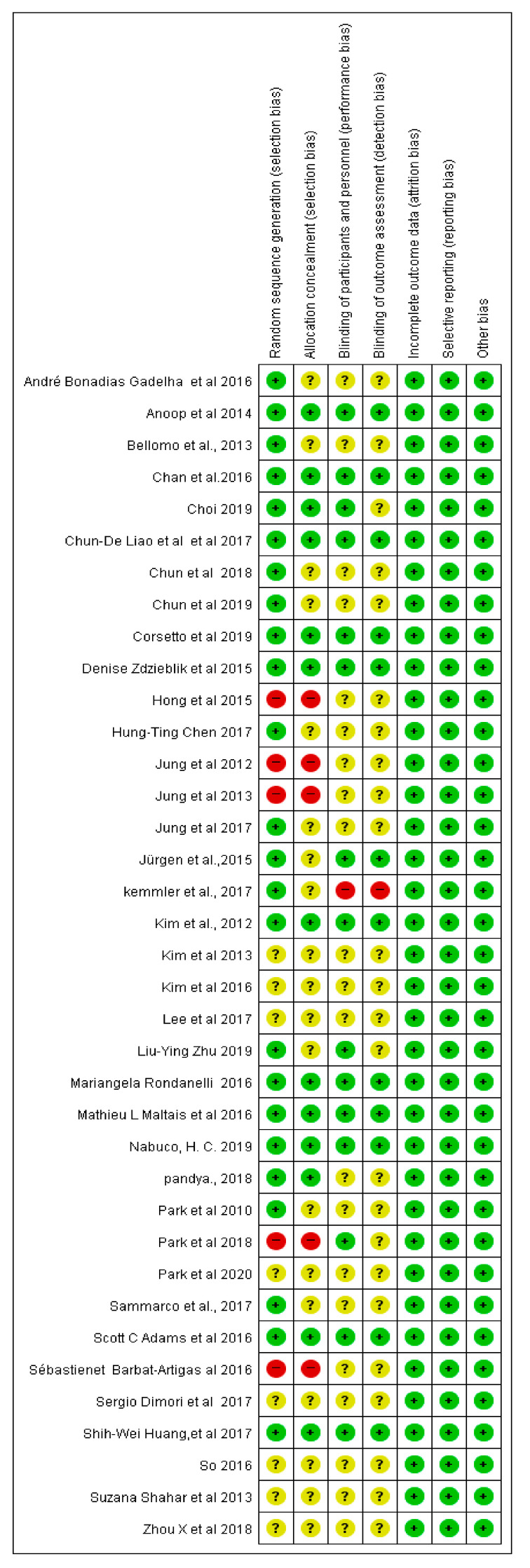
Risk-of-bias summary of systematic review on diagnostic tools and intervention methods for Sacopenia.

**Table 1 healthcare-10-00199-t001:** Characteristics of studies included in our systematic review.

Author (Year)	Sarcopenia Screening Measures	Participant Characteristics	Interventions (Combined_A_B, RET, AET, N, and Health Education)	Primary Outcomes
1. So (2016) [14]	BMI/body fat% ≥ 35% ASM < 5.14 kg/m^2^	SO (F) EG (*n* = 10, age = 67.9 years) CG (*n* = 10, age = 69.0 years)	Combined_A (complex: RET + AET) 12 weeks, 3 times per week	Body composition Fat (−)/FFM (+)/ASM (+)
2. Chun et al. (2019) [15]	Obesity: body fat ≥ 30% AWGS DXA	SO (F) A3-G = AEG (*n* = 14, age = 74.7 years) A5-G = AEG (*n* = 12, age = 72.7 years) R3-G = RET (*n* = 17, age = 74.2 years) R5-G = RET (*n* = 13, age = 73.8 years) CG (*n* = 18, age = 72.4 years)	Combined_A (complex: RET + AET) 6 weeks, 3 times per week 16 weeks, 5 times per week	Functional fitness test Chair stand test (+) Arm curl test (+) Chair sit and reach (+) Back scratch test (+) 2-min step test (+) 2.44 m Up-and-Go test (−)
3. Chun et al. (2018) [16]	Obesity: body fat ≥ 30% AWGS DXA	SO (F) A3-G = AEG (*n* = 14, age = 74.7 years) A5-G = AEG (*n* = 12, age = 72.7 years) R3-G = RET (*n* = 17, age = 74.2 years) R5-G = RET (*n* = 13, age = 73.8 years) CG (*n* = 18, age = 72.4 years)	Combined_A (complex: RET + AET) 16 weeks, 3 times per week 16 weeks, 5 times per week	Inflammatory factors CRP (−) ILs (+) Adiponectin (+)
4. Jung et al. (2017) [17]	ASM < 5.4 kg/m^2^	S (F) SG = EG (*n* = 12, age = 74.42 years) CG (*n* = 12, age = 73.58 years)	Combined_A (complex: RET + AET) 12 weeks, 3 times per week	Daily living fitness (+)
5. Park et al. (2018) [18]	AWGS	S (F and M) SG = EG (*n* = 22, age = 72 years) CG (*n* = 21, age = 72 years)	Combined_A (complex: RET + AET) 12 weeks, 3 times per weeks; total = 36 Self-efficacy education related motivation	Parameter muscle ASM (+)/-SMI (+)/-Handgrip strength (+)
6. Lee et al. (2017) [19]	ASM < 5.4 kg/m^2^	S (F) SG = EG (*n* = 12, age = 74.42 years) CG (*n* = 12/age = 73.58 years)	Combined_A (complex: RET + AET) 12 weeks, 3 times per week; total = 36	Ambulatory abilities (−) An ordinary pace (−) A quick pace (−)
7. Jung et al. (2012) [20]	BMI/body fat percentage ≥ 35% ASM < 5.14 kg/m^2^ DXA (Technical Insights, Inc.; Worthington, OH, USA)	SO (F) SOG = EG (*n* = 9, age = 74.10 years) NOG (*n* = 12, age = 74.82 years)	Combined_A (complex: RET + AET) 16 weeks, 3 times per week; total = 48	Body composition -Fat FFM (arms + legs + trunk) ASM (arms + legs)
8. Hong et al. (2015) [21]	DXA (Lunar iDXA, GE Healthcare Technologies, Chicago, IL, USA) Muscle index (limb muscle mass/weight) × 100 < 25.1%	S (F) SG = EG (*n* = 10, age = 80.50 years) CG (*n* = 12, age = 78.0 years)	Combined_A (complex: RET + AET) 12 weeks, 3 times per week; total = 30–50 min	Body composition Weight (+)/Percentage fat (−)/TMM (+)/SMI-related factors Upper Limb (+)/Lower Limb (+)/-ASM (+)/-SMI (+)
9. Park et al. (2020) [22]	<20 kg	S (F) SG = EG (*n* = 8, age = 8 years) CG (*n* = 8, age = 8 years)	Combined_A (complex: RET + AET) 15 weeks, 3 times per week; total = 60 min	Body composition Weight (+)/Percentage fat (−)/SMM (+)/WHR (−)
10. Jung et al. (2013) [23]	BMI/body fat percentage ≥ 35% ASM < 5.14 kg/m^2^	SO (F) SOG = EG (*n* = 9, age = 74.10 years) NOG (*n* = 12, age = 74.82 years)	Combined_A (complex: RET + AET) 16 weeks, 3 times per week; total = 48/50–60 min	Body composition Fat/FFM (arms + legs + trunk)/ASM (arms + legs)
11. Park et al. (2010) [24]	lean mass index (appendicular lean mass in kg/height in meters)/ASM = 27.3 kg	S (F) SG = EG (*n* = 10, age = 80.3 years) CG (*n* = 10, age = 81.2 years)	Combined_A (complex: RET + AET) 12 weeks, 3 times per week; total = 36/60 min	Weight (+)/Percentage fat (−)/Lean body mass 2-min step/Arm curl/chair sit and reach Back scratch/8 Foot Up-and-Go test
12. Huang et al. (2017) [25]	SMI < 27.6% BF > 30%	SO (F) SG = RET (*n* = 10, age = 80.3 years) CG (*n* = 10, age = 81.2 years)	RET = Elastic band resistance training 12 weeks, 3 times per weeks; total = 36/60 min	Body composition Percentage fat (−) -TMM (+)
13. Adams et al. (2016) [26]	SMI < 27.6%	S (F) RET (*n* = 64, age = 48.8 years) AET (*n* = 66, age = 48.9 years) CG (*n* = 70, age = 47.4 years)	RET (resistance exercise training) AET (aerobic exercise training) CG/4 weeks, 3 times per week; total = 36/60 min	RET (SMI (kg/m^2^) (+) AET (SMI (kg/m^2^) (+) CG (SMI (kg/m^2^) (+)
14. Balachandran et al. (2014) [27]	Obesity: BMI > 30 SMI Men < 10.76/m^2^ Women < 6.76/m^2^ Grip strength Men < 30 kg/Women < 20 kg	SO (F and M) SG = EG (*n* = 8, age = 71.6 years) CG = hypertrophy (*n* = 9, age = 71 years)	Combined_A (complex: RET + AET) 15 weeks, 2 times per week; total = 40–45 min	Body function SPPB modified (+)
15. Barbat-Artiga et al. (2016) [28]	BMI ≥ 30 kg/m^2^ Body fat ≥ 40%	SO (F) SG = EG (*n* = 50, age = 57 years) CG (*n* = 50, age = 51 years)	Combined_B1(AET + dietary plan) 3 weeks, 6 times per weeks; total = 60 min	Body composition
16. Gadelha et al. (2016) [29]	AFFM	SO (F) SG = EG (*n* = 64, age = 66.79 years) CG (*n* = 64, age = 67.27 years)	RET 24 weeks, 3 times per week; total = 60 min	Body composition Body mass (−)/BMI (−)/FFM (−) Relative FFM (−)/AFFM (−)
17. Zdzieblik et al. (2015) [30]	Handheld dynamometer Handgrip strength (<32 kg) DXA: measurement of muscle mass for sarcopenia classes I and II	S (M) EG (*n* = 26, age = 72.2 years) CG (*n* = 27, age = 72.2 years)	RET (resistance training program) 12 weeks, 3 times per week; total = 60 min Protein supplementation: collagen peptides/d Dietary intake	Body composition Body mass (−)/BMI (−)/FFM (−) Relative FFM (−) AFFM (−)
18. Zhou et al. (2018) [31]	AWGS	SO (M) EA + AA (*n* = 23, age = 70.35 years) AA (*n* = 25, age = 68.8 years)	EA (electrical acupuncture): orally twice per day for 28 weeks AA: every 3 days, for 12 weeks	Body composition BFP (−)
19. Zhu et al. (2019) [32]	AWGS	S (F and M) RET + nutrition (*n* = 36, age = 74.8 years) RET (*n* = 40, age = 74.5 years)	RET (resistance training program) 24 weeks, 2 times per week; total = 60 min The nutrition supplement consisted of two sachets of Ensure NutriVigor (Abbot Nutritional Products, Abbot Park, IL, United States) daily from baseline to 12 weeks. Each sachet (54.1 g powder) contains 231 calories, 8.61 g protein, 1.21 g	RET + Nutri RET Gait speed
20. Shahar et al. (2013) [33]	SMI Men < 10.75/m^2^ Women < 6.75/m^2^	S (F and M) EXR (AET + RET, *n* = 19, age = 69.74 years), PrG (*n* = 15, age = 65.93 years) EXR + PrG (*n* = 15, age = 65.20 years) CG (*n* = 16, age = 67.25 years)	Combined_A (complex: RET + AET) 12 weeks, 2 times per week; total = 60 min Protein: 1 soy protein drink a day (23% protein, 0.8% fat, 0.3% carbohydrates)	Body composition Weight (+)/BMI (+) Body fat (+)/FFM (+) Muscle (+)/Body water (+)
21. Maltais et al. (2016) [34]	Lean mass index < 10.75 kg	SO (M) RET (*n* = 8, age = 64 years) RET + dairy shake (*n* = 8, age = 8 years) Control (*n* = 10, age = 64 years)	RET (resistance training program) 16 weeks, 3 times per week; total = 60 min Control: Rice milk was used as a control supplement. Dairy shake: To attain the stipulated amount of protein, milk powder was added to 1% chocolate cow milk. Non-dairy protein shake: This consisted of a chocolate soy beverage to which commercial essential amino acids were added.	Body composition
22. Kim et al. (2016) [35]	Body fat ≥ 32% SMI < 5.67 kg/m^2^ Grip strength < 17.0 kg Walking speed < 1.0 m/s	SO (F) Exercise + nutrition (*n* = 36, age = 80.9 years) Exercise (*n* = 35, age = 81.4 years) Nutrition (*n* = 34, age = 81.2 years) Health education (CG, *n* = 34, age = 81.1 years)	Combined_A (complex: RET + AET) 12 weeks, 2 times per week; total = 60 min Nutrition (20 mg vitamin D, every 2 weeks taken daily with water for 3 months) Amino acid supplementation/Tea	Muscle mass Bodyweight/-ASM
23. Kim et al. (2013) [36]	Body fat percent ≥ 32% SMI < 5.67 kg/m^2^ Grip strength < 17.0 kg Walking speed < 1.0 m/s	S (F) EX + TC (*n* = 32, age = 81.1 years) Ex (*n* = 32, age = 79.6 years) TC (*n* = 32, age = 80.0 years) HE (*n* = 32, age = 80.2 years)	Combined_A (complex:RET + AET) 12 weeks, 2 times per week; total = 60 min Nutrition (20 mg vitamin D, every 2 weeks taken daily with water for 3 months.) Amino acid supplementation/Tea catechins (every 2 weeks, 1 bottle to drink daily for 3 months). Health education	Body composition Body mass Appendicular skeletal muscle Leg muscle mass
24. Liao et al. (2017) [37]	TSM SMI < 27.6% BF > 30%	SO (F) RET (*n* = 25, age = 66.39 years) Control (*n* = 21, age = 68.42 years)	RET (resistance training program) 12 weeks, 3 times per week; total = 45–55 min	Body composition (+)
25. Dimori et al. (2018) [38]	EWGSOP algorithm	S (F and M) Nutrition (*n* = 17, age = 86.5 years) RET + AET + Nutrition (*n* = 22, age = 85.6 years)	Combined_B (complex: RET + AET + nutrition) 48 weeks, 3 times per week; total = 40 min	Body composition -BIA/SMM/SMI
26. Chen et al. (2017) [39]	ASM Men ≤ 32.5% Women ≤ 25.7% BMI ≥ 25 kg/m^2^	SO (F and M) RET (*n* = 15, age = 68.9 years) AET (*n* = 15, age = 69.3 years) Combination (*n* = 15, age = 68.5 years) Control (*n* = 15, age = 68.6 years)	RET 8 weeks, 2 times per week; total = 60 min AET 8 weeks, 2 times per week; total = 60 min Combined_A (complex: RET + AET) 8 weeks, 2 times per week; total = 60 min	Body composition Weight/SMM/ASM/weight BFM/BMI/BFP/VFA
27. Rondanelli et al. (2016) [40]	Relative muscle mass Men = 7.26 kg/m^2^ Women = 5.5 kg/m^2^	S (F and M) RET + dietary G (*n* = 69, age = 80.77 years) Placebo (*n* = 6, age = 80.21 years) Food intake was based on a balanced diet.	RET: Physical activity 20-min exercise sessions daily, 5 times/week for 12 weeks.	Physical activity
28. Nabuco et al.(2019) [41]	SO was defined as a body fat mass ≥35% combined with ALST < 15.02 kg, assessed by DXA	SO (F) Whey + RET + Placebo (*n* = 22, age = 68.22) Placebo + RET + WHY (*n* = 21, age = 69.2) Placebo + RET + Placebo (*n* = 23, age = 70.1)	Resistance training program 16 weeks, 3 times per week; total = 80 min	Muscle mass
29. Corsetto et al. (2019) [42]	Relative muscle mass Men < 7.26 kg/m^2^ Women < 5.5 kg/m^2^	S (F and M) Supplement (*n* = 69, age = 80.77 years) Placebo (*n* = 61, age = 80.21 years)	Resistance training program 12 weeks, 5 times per week	Plasma fatty acids and oxidative stress markers of physical activity and amino acid supplementation
30. Choi et al. (2020) [43]	SMI < 30.52%	S (M) SG = EG (*n* = 23, age = 73.09 ± 1.69 years), SCG = CG (*n* = 24, age = 68.46 ± 2.20 years)	Combined_C1 (exercise + education) Exer: 12 weeks, 2–3 times per week Edu: 5 times a week, 12 weeks (1st: 55 min, 3rd: 60 min, 5th: 60 min, 9th: 60 min, 12th: 55 min)	SSEB (+)
31. Bauer et al. (2015) [44]	SPPB = 4–9, 20 kg/m^2^ < BMI < 30 kg/m^2^ SMI Men < 37% Women < 28%	S (F and M) SG = NT (*n* = 184, age = 77.3 ± 6.7 years) CG (*n* = 196, age = 78.1 ± 7.0 years)	NT: 13 weeks, 7 times per week; total = 182 min	Muscle strength Handgrip strength (kg) (+)/-SPPB (+) Function Chair stand time (s) (+), Balance test, -Gait speed (m/s) (−) Serum 25-hydroxyvitamin D (nmol/L) (+) Serum IGF-1 (ug/L) (+)
32. Chan et al. (2017) [45]	CSHA-CFS score 3–6	S (F and M) SG = EG (*n* = 143, age = 71.6 ± 4.28 years) SG = Control Group (*n* = 146, age = 71.3 ± 4.54 years)	EG = Combine F (RET + edu + psychotherapy) 12 weeks, RET 2 times per week + edu once for 2 months, psychotherapy once per month Control = Combine C1 2 weeks, RET 2 times per week + edu once for 2 months.	Body composition
33. Pandya (2019) [46]	Low muscle mass, low gait speed (<0.8 m/s in the 4-min walking test), low muscular strength (grip strength < 20 kg)	SO (F) Yoga = EG (pre: *n* = 788, age = 64.01 ± 3.86 years; post: *n* = 703, age = 73.21 ± 2.98 years) Control (pre: *n* = 788, age = 63.29 ± 4.01 years; post: *n* = 703, age = 72.33 ± 3.7 years)	Yoga Once a week for 480 weeks	DGI (+) Manual muscle test-8 (+)/Chair stand, arm curl (+) 6-min walk (+)/2-min step (+)/Chair sit and reach (inches) (+) Back scratch (inches) (+)/8 foot Up-and-Go test (s) (−)
34. Kim et al. (2012) [47]	Skeletal muscle mass/height^2^ < 6.42 kg/m^2^ and knee extension strength < 1.01 Nm/kg Skeletal muscle mass/height^2^ < 6.42 kg/m^2^ and usual walking speed < 1.22 m/s BMI < 22.0 kg/m^2^ and knee extension strength < 1.01 Nm/kg BMI < 22.0 kg/m^2^ and usual walking speed < 1.22 m/s	S (F) Combine 1 (*n* = 38, age = 79.5 ± 2.9 years) Exercise (*n* = 39, age = 79 ± 2.9 years) Nutrition (*n* = 39, age = 79.2 ± 2.8 years) Health education (*n* = 39, age = 78.7 ± 2.8 years)	Combine 1 (RET + NT) 12 weeks, twice per week. Total = 60 min, twice a day. The nutrition supplement consisted of 3 g amino acid × 2. RET 12 weeks, 2 times per week; total = 60 min twice a day Nutrition—twice a day The nutrition supplement consisted of 3 g amino acid × 2. Health education Once a month	Muscle mass (kg) (+) ASM (kg) (+))/LMM (+) Leg muscle mass (kg)(+)/BMI (−) Usual walking speed (m/s) (+) Maximum walking speed (m/s) (+) Knee extension strength (Nm) (+)
35.Sammarco et al. (2017) [48]	Fat mass > 34.8%, Fat-free mass < 90%, BMI > 25 kg/m^2^	SO (F) Nutrition (*n* = 9, X) Control (*n* = 9, X)	Nutrition 16 weeks, once a day. The nutrition supplement consisted of low calorie high protein diet 15 g protein + 1.2–1.4 g/kg body weight reference protein Cont Low calorie high protein diet 15 g protein + placebo	Body composition Weight (kg) (−)/-FFM (kg) (+) FAT (kg) (−)/-FAT (%) (−) REE (kcal/day) (−) REE/FFM (kcal/kg) (−)
36. Kemmler et al. (2017) [49]	FNIH and EWGSOP SMI < 0.789 Obesity: body fat > 27%	SO (M) WB + EMS + nutrition (*n* = 33, age = 77.1 ± 4.3 years) Nutrition (*n* = 33, age = 78.1 ± 5.1 years) Control (*n* = 33, age = 76.9 ± 5.1 years)	WB_EMS + Nutri -1.5 times per month + 1.7–1.8 g/kg per day body mass (100 g/day, 80 g protein, 9 g high L-leucine, 57 g essential amino acid, 2.8% fat, 6.4% carbohydrates) Placebo 800 IU cholecalciferol	Sarcopenia Z-score Total body fat (%) SMI/Handgrip strength (kg)
37. Bellomo et al. (2013) [50]	Centers for Disease Control and Prevention Muscle mass index = muscle mass/height less than two deviations below the mean of a young reference population	S (M) Sensorimotor training (*n* = 10) Exercise (*n* = 10) Vibratory (*n* = 10) Control (*n* = 10)	Sensorimotor training 12 weeks, twice a week. Total = 20 min Exercise 12 weeks, twice a week. Vibratory 12 weeks Weeks 1–8: once a week Weeks 9–12: twice a week Total = 15 min Control	Leg extension 90 right limb Leg extension 90 bilateral limb Eye open and close sway area Eye open ellipse surface (mm) Eye close ellipse surface (mm) Half-step length (cm) Step width (cm)

Combined_A (combined resistance exercise and aerobic exercise), Combined_B (complex: RET + AET + nutrition), RET: resistance training program, AET: aerobic training program, HE: health education, BMI: body mass index, ASM: appendicular skeletal muscle mass, SO: sarcopenic obesity, EG: experiment group, CG: control group, IGF-1: insulin growth factor-1, AWGS: Asian Working Group for Sarcopenia, DXA: dual-energy X-ray absorptiometry, A3-G: aerobic exercise group 3 days a week, AEG: aerobic exercise group, A5-G: aerobic exercise group 5 days a week, R3: resistance exercise group 3 days a week, R5: resistance exercise group 5 days a week, CRP: C-reactive protein, ILs: Inflammatory cytokines, SG: sarcopenia group, F: female, IL6: interleukin 6, SPPB: short physical performance battery, SMI: skeletal muscle mass index, SOG: sarcopenia obesity group, NOG: non-obesity group, TMM: trunk muscle mass, WHR: waist circumference/hip circumference, SMM: skeletal muscle mass, BF: body fat, AFFM: appendicular fat-free mass, FFM: fat-free mass, EA: electrical acupuncture, AA: acupuncture, BFP: body fat percentage, IU: international units, PrG: protein intake group, Nutri: nutrition, Edu: education, TC: tea catechins, BIA: bio-electrical impedance analysis, EWGSOP: European Working Group on Sarcopenia, VFA: volatile fatty acids, ALST: appendicular lean soft tissue, SSEB: simplified surface energy balance, LMM: low muscle mass, REE: resting energy expenditure, FNIH: the Foundation of the National Institutes of Health, M: male, +: positive effect, −: negative effect.

## Data Availability

Not applicable.

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
