# Peer review of "Systematic Review of Diagnostic Tools and Interventions for Sarcopenia"

_healthcare, 2022, doi:10.3390/healthcare10020199_

Round 1
Reviewer 1 Report
In the submitted manuscript, Cheong et al, present a systematic review on diagnostic tools and intervention in sarcopenia treatment. Criticisms have been identified, mainly regarding the source of information, data presentation and literature reporting. This Reviewer recommends major revision.
In particular:
- the authors shoud discuss better in the introduction/discussion section how the available body of evidence presented in other systematic reviews about sarcopenia (119 in 2021 searching on PubMed) is concordant or discordant with the analysis presented in the manuscript. Which type of topic is presented in other systematic reviews on sarcopenia? Do other research groups reported other systematic reviews about diagnostic tools and interventions? Are the data consistent? How should the research presented in the manuscript consist in an actual advantage for researcher studying sarcopenia in a clinical environment? Pleas discuss throroughly.
- Section "Material and methods", subsection 2.1 "Information sources". The authors should justify the reason why they chose to retrieve data from many different sources including "grey papers" and Korean language sources that should not be available to the international community. Moreover, the authors should ensure and provide evidence that all the information retrieved are part of peer-reviewed literature. This should not be the case of "grey papers" that should not be included in the analysis at all. In the opinion of this Reviewer, a systematic information retrieval based on PubMed could represent a source available to the whole scientific community and is based only on peer-reviewed works. The authors could build a graph resuming the source of the 37 articles that were finally included in the analysis to allow the comprehension by the reader about the source of information.
- Figure1. Please insert a figure description following the caption.
- Section "3. Results"; subsections "3.2 Sarcopenia screening measures" and "3.5 Interventions". These sections lacks of information that could be very interesting to the readership. The authors could build one or more graphs to resume the distribution of diagnosis tools and intervention tools across the articles analysed in this study. Moreover, the different tools (diagnosis and intervention) are not discussed about the quality of results that give as outcome. Which kind of diagnostic tools gives more precise results (as lower standard deviation) and which kind of interevention is able to better alleviate the sarcopenic situation of the patients? It is true that the outcomes registered by the 37 articles are different but some overlap is present. Is/Are there one or more outcome parameters that are more frequently used and give more consistent results? How this difference could impact on research and clinical practice?
- Figure 2 and Figure 3: please insert a figure legend
- For all the figures: please provide the images in a better quality (at least 300 dpi) and a sentence to state that these figures are original.
Author Response
- In previous previous studies, there was a topic on the prevention and treatment methods of exercise therapy in relation to sacopenia. In addition, there is a systematic review of health indicators, and Reference No. 52 has been added to this section.
- There are no systematic review articles discussing the clarity of diagnostic tools. Only the Report of the International Sarcopenia Initiative (EWGSOP and IWGS) exists.
- The data were reviewed by two investigators, which were then reviewed by a third investigator.
- The discussion section has been modified as follows. Finally, in South Korea, 560 adults over the age of 65 were followed for 6 years, and the risk of death was 4.1 times higher in male sarcopenia patients with insufficient muscle mass and strength. Slow walking increased the risk of death five times. There are many reports that cancer patients with sarcopenia have a shorter survival time and a worse prognosis, including recurrence. However, in the case of Korea, sarcopenia is not classified as a disease yet, and prevention or diagnosis is not clear. Therefore, if a clinician can suspect sarcopenia when discussing the ambiguity of the sarcopenia diagnostic tool and treating the elderly in clinical practice, it will be possible to provide various options to the patient. Therefore, the results of this study are expected to provide various clinical advantages for future sacopenia research.
- In Korea, a gray thesis refers to a case in which one of the dissertations has not been published. Accordingly, it was written to see the trend of the research topic. It was not included in the analysis.
- Changed to PRISMA flow chart of Paper search for systematic review on diagnostic tools and intervention methods for Sacopenia.
- This part was not repeated because there was a diagram in table1. Please consider the advantages. There was no evidence that aerobic exercise and strength exercise were more effective than aerobic exercise and strength exercise, as confirmed in the 3.5 intervention technique. All I said was that they were all positively effective and statistically significant. As a result, it seems necessary to conduct a meta-analysis that can compare statistical figures to confirm the results. This part was written in the discussion. Notably, most studies have been conducted in women [55], suggesting that sarcopenia has sex specificity. This finding is in agreement with the finding of So in 2016 [14], who reported a higher risk of sarcopenia in women than in men. The clinical intervention was circuit exercise combining alternating resistance training and aerobic exercise; resistance training was used to increase muscle strength and aerobic exercise to decrease fat. This is consistent with the recent findings reported by Yin et al. in 2020 [12], who stated that the use of circuit exercise was increasing worldwide as an intervention for sarcopenia and other diseases in older adults. In addition, Rodrigues-Krause et al in 2021 [56] stated that physical interventions for older adults should be structured and include aerobic and strength training, while ensuring that these adults can enjoy performing them by themselves [57]. This may be the basis for the fact that the interventions used in the selected studies mainly aimed to decrease fat using aerobic exercise such as dancing. However, it is not possible to confirm the optimal intervention for comparative advantages. To this end, we propose a study be conducted to confirm priorities of interventions through comparative analyses of future interventions. In addition, strength, muscle mass, fat, body fat, and gait performance were mainly used as tools to measure the effectiveness of the intervention, but there was no consistency in the use of diagnostic tools. There was a commonality between measuring body composition and measuring physical strength, but the reference points were not consistent. This result supports the necessity to discuss not only the consistency of diagnostic tools but also the uniformity of outcome measurements because the evaluation criteria for sarcopenia are diverse.
- I wrote both figure 2 and figure3 legends. thank you. Thank you for your kind and attentive review and for helping to make this thesis better in many parts.
- RoB assessment was performed for the 37 included studies using Review Manager 5.4.1 (Figs. 2 and 3).
thank you.
Reviewer 2 Report
Sarcopenia is a progressive and generalized skeletal muscle disorder involving the loss of muscle mass, which affects physical activity levels. Lower physical activity levels further contribute to muscle shrinkage.
In general, sarcopenia is diagnosed by evaluating muscle mass, muscle strength, and muscle function or performance. All definitions use at least two parameters but different cutoff points lead to a lack of standardization and poor application of these definitions in clinical practice.
In this manuscript, the authors systematically and comprehensively reviewed several diagnostic and evaluation criteria and interventions for sarcopenia in the recent 10 years across published articles. According to the authors’ findings, no clear standard tools for the diagnosis of sarcopenia as various criteria were applied, currently.
It is a good review article, useful for the effort to harmonize definitions and diagnostic criteria of sarcopenia, also provide basic information for rehabilitation treatment.
No comment.
Author Response
Thank you for your kind and warm review. Thank you again for giving me a good review and giving me the opportunity to post this.
Reviewer 3 Report
I would like to say that I am very thankful to have the opportunity to read this study. The suggestions given in this document are intended to improve your work.
General comment:
I think the work is well performed and well written. I would like to make a few comments, and I would also suggest checking minor mistakes in the translation.
Introduction section:
- The authors demonstrate great knowledge of the subject. Perhaps they could expand the introduction to better explain its characteristics, impact...
Methods section:
- Although the authors claim to have used the 2020 flow diagram, I suggest them to check this figure again. In addition, please enlarge it to make it more legible.
- http://prisma-statement.org/PRISMAStatement/FlowDiagram
Discussion section:
- The discussion is scarce. Please elaborate on the variables analyzed and compare your results with other works to give it depth.
- 2 It should be renamed as Limitations and Future Lines.
Author Response
Thank you for your kind and warm review. Thank you again for giving me a good review and giving me the opportunity to post this.
General Comment: Thank you for your kind and thoughtful review. I entrusted the translation and proofreading once again.
Introduction section:
thank you. Thanks again for the thoughtful review.
Methods section:
Changed to 2020 flow chart.
Thank you for your kind review.
discussion section
Edited. thank you.
The picture has a higher resolution.
Renamed to Yes Limitations and Future Lines.
Really, thank you. Thank you for your kind and warm review, and for helping to make this thesis better in many parts.
Round 2
Reviewer 1 Report
Dear Authors,
you have accomplished the point raised in the previous review report.